# Identification and Analysis of Genes Related to Testicular Size in 14-Day-Old Piglets

**DOI:** 10.3390/ani14010172

**Published:** 2024-01-04

**Authors:** Yunjiao Zhao, Liangzhi Zhang, Lei Wang, Jianbo Zhang, Wenjuan Shen, Yuhong Ma, Chengxiang Ding, Guofang Wu

**Affiliations:** 1Plateau Livestock Genetic Resources Protection and Innovative Utilization Key Laboratory of Qinghai Province, Qinghai Academy of Animal and Veterinary Medicine, Qinghai University, Xining 810016, China; zhaoyunjiao2021@163.com (Y.Z.); wanglei19831002@gmail.com (L.W.); zhangjb9122@163.com (J.Z.); shenwenjuan0518@163.com (W.S.); 2004990028@qhu.edu.cn (Y.M.); 2011990016@qhu.edu.cn (C.D.); 2Qinghai Provincial Key Laboratory of Animal Ecological Genomics, Northwest Institute of Plateau Biology, Xining 810008, China; lzzhang@nwipb.cas.cn

**Keywords:** testis, early development, candidate genes, RNA-seq, pig

## Abstract

**Simple Summary:**

The reproductive performance of pigs can vary due to the influence of genetic factors, and this is aggravated by environmental factors. Multiple complex factors affect the difference in the reproductive performance of boars in adulthood. This study examined those genetic differences in young piglets, at a time when they are less affected by the environment. Testicular tissue samples were used to explore candidate genes and related pathways affecting testicular size during early development. This experiment into the reproductive performance of pig breeds using the RNA-Seq analysis of testicular tissue with early developmental differences can provide a theoretical basis for breeding by determining key candidate genes that control testicular predevelopment. The data will be used for subsequent research into the genetic mechanism of boars, which is important for early breeding and improvement.

**Abstract:**

The RNA-Seq technology was used to screen the key genes that affect the early development of the testes of Duroc × Landrace × Yorkshire piglets, to determine the regulatory pathway and provide reference for subsequent reproductive performance research, breeding, and other production practices. This study selected 14-day-old Duroc × Landrace × Yorkshire piglets as the trial animals. Testes from piglets with similar weights and no pathological changes were divided into small testis (ST) and large testis (LT) groups, and the RNA-Seq screening of differentially expressed genes (DEGs) was performed to find candidate genes and regulatory pathways related to early testicular development. The results show that 570 DEGs were found in the ST and LT groups, with 281 upregulated and 289 downregulated. The DEGs were mainly enriched on 47 gene ontology (GO) functional items. The Kyoto encyclopedia of genes and genotypes (KEGG) enrichment analysis found that there were 44 significantly enriched KEGG signal pathways, and the regulation of testicular development mainly focused on the arachidonic acid metabolism, Wnt signaling pathway and GnRH secretion pathways. The *PTGES*, *SFRP1*, *SPP1*, *PLA2G4E*, *KCNJ5*, *PTGS2*, and *HCN1* genes were found to be as closely related to the testicular development of these Duroc × Landrace × Yorkshire piglets, and the differential gene expression was consistent with the real-time quantitative reverse transcription PCR (real-time qRT-PCR) validation results. This study was validated by high-throughput sequencing analysis and real-time qRT-PCR, and showed that the *PTGES*, *SFRP1*, *SPP1*, *PLA2G4E*, *KCNJ5*, *PTGS2*, and *HCN1* genes may be involved in the regulation of germ cell development, spermatogenesis and semen traits. These should be further studied as candidate genes for early testicular development and reproductive trait regulation in boars.

## 1. Introduction

The testis is an important organ, which has the function of producing sperm, secreting male hormones, accommodating the growth and development of germ cells, maintaining normal physiological structure and physiological function, and the positive regulation of reproductive activities. Testicular size is generally based on the scrotal circumference (SC), which is one of the main indicators of genetic improvement in breeding animals [1]. Testicular development includes the proliferation, differentiation, growth and function of germ cells, which is a complex and delicate process. Studies have shown that testicular size has positive effects on ejaculation volume, sperm density and sperm motility in different male mammals, while the rate of sperm malformation is inversely correlated [2]. Testicular size is therefore an important phenotype to measure the reproductive performance of male animals.

In recent years, RNA-Seq technology has received widespread attention, and many discoveries have been made in the field of reproduction. Zhang et al. [3] compared testicular development in 30- and 120-day-old cooperative and long-white boars, and found that testicular development and spermatogenesis played a crucial role in the early maturation of pigs, through various signaling pathways and gene regulation, and screened relevant long noncoding RNAs (lncRNAs) that may be involved in testicle development and spermatogenesis, including LOC102166140, LOC110259451 and MSTRG.15011.2, as well as mRNA including *PDCL2*, *HSD17B4*, *SHCBP1L*, *CYP21A2* and *SPATA3*. Ma et al. [4] sequenced endogenous PIWI-interacting RNAs (piRNAs) expressed in the testicular tissues at 2, 3, 6 and 12 months, and observed that the composition of piRNA was associated with testosterone (TE) in the 2- and 3-month-old testes, while the testes at 6 and 12 months of age were mainly composed of gene-derived and intergenic piRNAs and their expression levels gradually increased during testicular development. More than half of the piRNA readings were mapped to several of the predicted piRNA clusters, including *CYP19A1*, *PRMT8*, *SUZ12*, *WWOX*, *SGSM1* and *MIF*, with the function of these genes mainly related to steroid generation and histone modification. Changes in piRNA composition and a wide range of expression patterns during sperm cell development have shown that small noncoding RNAs (ncRNAs) may be responsible not only for transposon inhibition, but also for the post-transcriptional regulation of several protein-coding genes necessary for normal spermatogenesis. Yang et al. [5] used the RNA-Seq datasets of five mammals, including pigs, cattle, sheep, humans and mice, to analyze the expression pattern of protein-coding genes in pig testicular tissues through intraspecific and interspecific comparisons and evaluate genes related to male reproduction in pigs, including their transcriptional characteristics and expression regulatory mechanisms. This study found 1210 candidate genes related to male reproduction in pigs, including 87 low-expression-conservation testis-specific genes (TSGs), 113 moderate-expression-conservation TSGs and 195 high-expression-conservation TSGs, and three molecular mechanisms that regulated their expression via an interaction between alternative splicing (AS), transcription factors (TFs), and TSGs. These findings provide new insights into gene expression profiles related to testicular development and function, and offer a rationale for further mechanistic studies.

How mRNA regulates the development of mammalian testis needs to be further studied in depth, so it is of great scientific significance to explore the influence of molecular regulation and candidate genes on the development and reproduction process of testicular tissue. This study aimed to improve the understanding of the changes of testicular gene expression during fetal testicular development and provide new insights into the mechanisms of early testicular development, which is crucial to the success of the pig industry. In this experiment, 100 testicular samples of 14-day-old piglets were collected, combined with phenotype data such as testicular weight and testicular circumference, and the samples were allocated to large testicular and small testicular groups. Histological examination and transcriptome sequencing were conducted to determine the pathways related to testicular development differences and to select relevant candidate genes to provide data support for subsequent reproductive performance studies and early breeding.

## 2. Materials and Methods

### 2.1. Test Animal Selection and Sample Collection

All the protocols followed in these experiments were approved by the Ethical Committee of the Qinghai Academy of Animal Science and Veterinary Medicine. The test animals were provided by Qinghai Yufu Animal Husbandry Development Co., Ltd., Qinghai, China, as 100 healthy 14-day-old Duroc × Landrace × Yorkshire piglets with similar weights and genetic backgrounds from the same breeding environment. After castration, testicular tissue, as well as testicular weight, length and width data, were collected. According to the SC distribution of all testicular samples, they were divided into a small testis (ST) group of 42 samples and a large testis (LT) group of 58 samples. Three testes with similar weights from healthy piglets were selected for follow-up experiments including histological observations, RNA-Seq sequencing and real-time qRT-PCR validation in each group where the ST group was the control group and the LT group was the experimental group. The testicular tissue was stored in 4% paraformaldehyde and liquid nitrogen to be made into paraffin sections to observe testicular development, where total RNA was extracted for transcriptome sequencing.

### 2.2. Materials and Methods

#### 2.2.1. Histological Testing

The testicular tissue of piglets was prepared for histological observation using hematoxylin–eosin (HE) staining.

(1) Dehydrated and transparent:

A move from a low concentration to a high concentration of alcohol as the dehydrating agent gradually removed the water in the tissue block, which was then placed in a transparent agent xylene and the alcohol in the tissue block was replaced with xylene.

(2) Paraffin embedding:

The transparent tissue block was placed in dissolved paraffin and in a dissolved wax box for insulation. After the paraffin was completely immersed in the tissue block for embedding, the container was prepared with the dissolved paraffin tissue block and the paraffin-soaked tissue block quickly clipped into it, and was then cooled and solidified into blocks.

(3) Slice and patch:

The embedded wax pieces were cut into thin slices, flattened in heated water, stuck to the slide, and dried in a 45 °C incubator.

(4) Dewaxing staining:

Before staining, the paraffin in the section was removed with xylene, then moved from a high concentration to a low concentration of alcohol, and finally into distilled water.

The HE staining process:

① The sections after immersion in distilled water were stained in hematoxylin solution for 30 s;

② Color separation in acid water and ammonia water each for 5 s;

③ Rinsed under running water for 1 h and then added with distilled water for 5 s;

④ Dehydrated in 70% and 90% alcohol for 10 min each;

⑤ Alcohol eosin staining solution was used for 2 to 3 min;

(5) Dehydration and transparent:

The stained sections were dehydrated with pure alcohol and then passed through xylene to make the sections transparent.

(6) Sealing:

Canadian gum was dropped on the slice and this was covered with cover slides. After the gum was slightly dry, the label was pasted on. Finally, microscopy was performed and the sections were scanned.

On each section, a convoluted seminiferous tubule with a round cross-section was selected to observe the germ cells and the seminiferous tubules.

#### 2.2.2. RNA-Seq

Total RNA was extracted from the testicular tissue samples of each group using the TaKaRa Mini BEST Universal RNA Extraction Kit (Takara Biomedical Technology (Beijing) Co., Ltd. Beijing, China) to detect the concentration, purity and integrity of the extracted RNA, which was saved for subsequent testing and sequencing. A Truseq TM RNA sample prep Kit was used to construct the library. The sequencing columns were analyzed statistically and for quality control using the Illumina Novaseq™ 6000 platform. TopHat2 (http://tophat.cbcb.umd.edu/ (accessed on 17 April 2022)) [6] software was used to obtain the Clean reads and reference genome sequence alignment analysis (http://asia.ensembl.org/Sus_scrofa/Info/Index (accessed on 7 April 2022)) was used to get the sequence information.

#### 2.2.3. Differential Expression Analysis and Differential Expression Genes (DEGs) Enrichment Analysis

The expression levels of genes were quantified using DEGseq software [7], and Transcripts Per Million Reads (TPM) was used as a standard to measure the expression levels for the experimental and control groups. Using RSEM [8], the read counts of each sample gene were obtained, the gene expression level was obtained and DEGs were identified between samples to study their function. The screening threshold of DEGs was |log_2_Fold change| ≥ 1 and *p*-value < 0.05, and the software GOatools (https://github.com/tanghaibao/GOatools (accessed on 8 May 2022)) [9] was used to perform GO enrichment analysis on the genes in the obtained gene set; when the *p* value (FDR) < 0.05, it was considered that there was significant enrichment of their GO function. R (v4.2.1, https://www.r-project.org (accessed on 10 May 2022)) was used to write a script to perform the Kyoto encyclopedia of genes and genomes (KEGG) enrichment analysis of genes in the gene set, and the calculation principle and visualization of the results referred to the GO function enrichment analysis standard.

#### 2.2.4. Real-Time qRT-PCR Validation

Seven DEGs were selected to verify the real-time qRT-PCR of the testicular samples and β-actin was used as the reference gene to test the accuracy of sequencing results and data analysis. According to the NCBI sequence information, the primers of the internal reference gene and the DEGs were designed by Primer Express v3.0 (Thermo Fisher Scientific, Waltham, MA, USA) and synthesized by Shenggong Biotechnology Co., Ltd. (Shanghai, China), with the gene names and primer information shown in Table 1. Using each sample RNA pool as a template, the TaKaRa PrimeScript™ RT reagent Kit with gDNA Eraser (Perfect Real Time) (Takara Biomedical Technology (Beijing) Co., Ltd. Beijing, China) kit was used for reverse transcription and amplification. The 2^−ΔΔCt^ method calculated the relative expressions of genes and the fold change.

### 2.3. Data Processing and Analysis

Excel 2019 (Microsoft, Redmond, WA, USA) was used to organize the basic data of the test, SPSS (v23.0, https://www.ibm.com/cn-zh/spss (accessed on 22 September 2022)) was used to perform a one-way analysis of variance (ANOVA) on the obtained test data, and the least significant difference method was used to analyze the significance of the differences in each treatment group.

The results were visualized using R 4.2.1 and the difference in ΔCt value was used to calculate the difference in gene expression as
Ct target gene − Ct internal reference gene = ΔCt.

## 3. Results

### 3.1. Histological Observation

Histological observations showed that the seminiferous tubules had no lumen, and the types of cells in the seminiferous tubules were mainly primitive germ cells, spermatogonia and Sertoli cells. Primitive germ cells were mostly located near the basement membrane of the seminiferous tubules, and their cell bodies were relatively small. The Sertoli cells in the seminiferous duct were arranged in large numbers in a columnar shape on the inner side of the basement membrane. A small amount of spermatogonia were distributed at the basement membrane, intermediate between the Sertoli cells. The Leydig cells had a large volume, clear cell boundaries and vacuolar structures due to fat droplets in the cytoplasm. The nucleus was large, circular and lightly stained. The diameter of the seminiferous tubules in the LT group at 59.01 ± 1.56 μm was significantly larger than that of the ST group at 49.55 ± 2.85 μm (*p* < 0.05), and the proliferation and development rates of Sertoli cells, Leydig cells and spermatogonia in the ST group lagged behind those in the LT group, as shown in Figure 1.

### 3.2. Quality Control Data Statistics and Sequence Alignment

The RNA-Seq of testicular tissue samples in the ST and LT groups obtained a total of 52.26 Gb of clean data; all samples were above 8.32 Gb, the Q30 base percentage was greater than 94.25, and the library quality and sequencing quality were high and worthy of further analysis. The alignment rate was between 96.01 and 96.45%, with the specific results shown in Table 2.

### 3.3. DEGs Screening

A total of 25,711 expressed genes were detected in this experiment, including 25,000 known genes and 711 new genes. A total of 62,181 expressed transcripts were present, including 46,998 that were known and 15,183 that were new. A total of 570 DEGs were found in the ST and LT groups, with 281 genes upregulated and 289 downregulated genes, shown in Figure 2.

Cluster analysis was performed on 570 selected DEGs, which showed the expression levels of DEGs in ST and LT groups and the differences in gene expression between ST and LT groups that was observed. In Figure 3, the related genes of the repeat in the two groups were clustered separately as the abscissa, and the two gene clusters were significantly separated by the upward and downward regulation of DEGs as the ordinate, indicating that there was a high intra-class correlation and the sample grouping was reasonable.

### 3.4. GO Function Enrichment Analysis of DEGs

The GO analysis of the 570 obtained DEGs found that DEGs were annotated on 47 GO function items. There were three categories, including biological process, of which the highest category was cellular process; cellular component; and the highest category was cell part and molecular function, with the highest proportion being binding, shown in Figure 4.

### 3.5. KEGG Function Enrichment Analysis Results of DEGs

The KEGG signaling pathway enrichment analysis found that DEGs were enriched in 270 KEGG signaling pathways, and the first 25 KEGG signaling pathways enriched are shown in Figure 5. These include immune pathways, mainly GnRH secretion and arachidonic acid metabolism, of which there are 14 pathways related to testicular predevelopment: steroid biosynthesis, Wnt signaling pathway, primary bile acid biosynthesis, GnRH signaling pathway, GnRH secretion, steroid hormone biosynthesis, arachidonic acid metabolism, cell cycle, mTOR signaling pathway, apoptosis, apoptosis–multiple species, necroptosis, cellular senescence and growth hormone synthesis, secretion and action.

To better reflect the interaction between these pathways, the interaction between the above 14 KEGG pathways and genes was analyzed in Figure 6 and the results show that the regulation of testis development was mainly concentrated in arachidonic acid metabolism, Wnt signaling and the GnRH secretion pathways. The *PTGES*, *SFRP1*, *SPP1*, *PLA2G4E*, *KCNJ5*, *PTGS2*, and *HCN1* genes were found to be closely related to the development of testes in Duroc × Landrace × Yorkshire piglets.

### 3.6. Real-Time qRT-PCR Verification Results

The real-time qRT-PCR results of seven genes are shown in Figure 7, and the expression of target genes between the two groups was consistent with the RNA-Seq results. The expression levels of PTGES, SPP1, PLA2G4E, KCNJ5, and PTGS2 were higher in the LT group, and SFRP1 and HCN1 were higher in the ST group.

## 4. Discussion

Testis development and spermatogenesis are complex biological processes regulated by different genes at different developmental stages, and gene expression is dynamic and stage-specific [10]. Some mRNAs have been found to have regulatory functions in testicular development and spermatogenesis. Chen et al. [11] explored the role of *SOX9* in mammalian reproductive regulation, which is significantly correlated with testicular weight, revealing for the first time that the polymorphism mutation and expression of the *SOX9* gene was significantly correlated with the pig reproduction trait, indicating the important role of the *SOX9* gene in testicular development. Zhang et al. [12] showed that *StAR-a* had a high expression in testicular tissue and *StAR* and *StAR-a* played important roles in male reproductive ability. Kaczmarek et al. [13] found that Ccdc33 encoded a specific peroxisome protein that was mainly expressed in male germ cells, and this protein played an important role in spermatogenesis. Zhang’s [7] GO and KEGG analysis of target genes of mRNA showed that most of them were rich in some common metabolic pathways, such as TGF-β, PI3K-Akt and Wnt, that played a functional role in testis development and spermatogenesis [14,15,16,17].

The Wnt signaling pathway played an important role in testis differentiation and spermatogenesis, and through these analyses, some candidate genes that may be associated with precocious puberty traits were identified. The testicular development-related pathways identified in this study included arachidonic acid metabolism, Wnt signaling pathway and GnRH secretion, and seven candidate genes closely related to the development of testes of Duroc × Landrace × Yorkshire piglets were screened, including *PTGES*, *SFRP1*, *SPP1*, *PLA2G4E*, *KCNJ5*, *PTGS2*, and *HCN1*.

Two to three weeks after birth is the peak period for the proliferation of testicular interstitial cells in piglets. Sertoli cells are the earliest somatic cells formed in the testes. During spermatogenesis, they not only provide nutrients and growth factors for various levels of germ cells, but also determine the size of the testes based on their number. The two important stages in testicular development are before sexual maturity, including the fetal stage and childhood stage, and after sexual maturity [18]. Before sexual maturity, the Sertoli cells, stromal cells and spermatogenic cells in testicular tissue are in a rapid proliferation and differentiation state, while after sexual maturity, the proliferation state enters the plateau phase. This experiment selected 14-day-old testicular tissue samples, and histological analysis showed that the seminiferous tubules were composed of a large number of Sertoli cells and a small amount of spermatogonia, with the number of Leydig cells relatively small and no true lumen at this time. The diameter of the spermatogenic duct of the boars in the LT group was greater than that of the ST group boars of the same age, and the development of Leydig, Sertoli and germ cells also differed, indicating that there were group differences in the tissue structure of testes at the same age.

The sequencing results show that compared with the ST group, *SFRP1* was expressed in the LT group, and its expression had an inhibitory effect on testicular development. Wong et al. [19] found that *SFRP1* was an important regulatory protein for sperm formation and the expression of *SFRP1* was strictly regulated in adult rat testes to control sperm adhesion and sperm release. The injection of *SFRP1* recombinant protein into testes delays sperm fertilization, accompanied by phosphorylated, focal adhesion kinase-Tyr (397) downregulation and ectoplasmic specialization (a testis-specific anchoring junction) and apical adhesion protein retention, highlighting the important role of *SFRP1* in regulating sperm fertilization through its effects on the focal adhesion kinase signaling pathway and the ectoplasmic specialization apical adhesion complex.

Components of the Wnt signaling pathway are involved in regulating cell morphology, proliferation, motility, and cell fate [20]. The gene *SFRP* is an extracellular inhibitor of Wnt signaling. Wnt/β-catenin signaling is required for the normal development of primordial germ cells, mediated by the secreted curve-associated protein (SFRP) family, including five different glycoproteins *SFRP1* to *SFRP5*, and this is the largest of the Wnt signaling pathways and acts by binding directly to Wnt ligands or curl receptors [21]. Fabijanovic et al. [22] showed that the expression of *SFRP1* in seminomas and mixed or nonseminomas was lower in atrophied tissues compared with benign tissues, and *SFRP1* appeared to be actively involved in the pathogenesis of primary testicular germ cell tumors, indicating that the downregulation of *SFRP1* was actively involved in the regulation of testicular development. The results of this experiment showed that *SFRP1* was negatively correlated with testicular size and development, which was similar to the regulatory conclusion of Fabijanovic, so in testicular development, *SFRP1* was inferred to be involved in the negative regulation of testicular development.

The genes *PTGES*, *SPP1*, *PLA2G4E*, *KCNJ5*, and *PTGS2* were highly expressed in the LT group and positively regulated testicular development. The key to testicular development depends primarily on Leydig cells, testicular Sertoli cells, and spermatogonia [23]. The *SPP1* protein is present in germ cells, support cells and Leydig cells within the seminiferous tubule [24]. Also known as osteopontin, it is an extracellular matrix protein that plays an important role in testicular tissues [25]. Stenhouse et al. [26] observed significant changes in *SPP1* expression in the fetal testes in pigs during late pregnancy, and *SPP1*, integrins and other ligands of the integrins receptor such as fibronectin were involved in the mechanisms involved in fetal testicular development. The genes *SPP1*, *KCNJ5*, and *HCN1* belong to the GnRH secretion signaling pathway, and there are few studies on their correlation with testicular development. The GnRH secretion signaling pathway has an important impact on testicular development. The GnRH neurons are the central part of the hypothalamic–pituitary–gonadal axis that regulates reproduction, and GnRH secretion is activated periodically during fetal development, postnatal “micro-puberty” (the negative feedback link between the hypothalamic–pituitary–gonadal axis is established in newborn and early infancy, but its inhibitory function is not mature; at this time, gonadotropin is highly secreted at pubertal levels, so it is called “micro-puberty”) and puberty [27,28]. During puberty, GnRH neurons are gradually reactivated, increasing pituitary gonadotropins and eventually producing gonadal steroids. Lund et al. [29] used RNA sequencing to analyze the transcriptome characteristics of specific GnRH neuronal progenitors and early post-mitosis GnRH neurons, and FGF8 signaling regulated these genes upstream during GnRH neuronal differentiation, explaining how GnRH neuronal transcriptome analysis can be used to infer the signaling pathways and gene regulatory networks involved in human GnRH neuronal development.

The gene *PTGES* encodes human microsomal PGE synthase 1 and catalyzes the conversion of PGH2 to PGE2, which plays a key role in the inflammatory response [30] and cell growth and transformation [31], and a study showed that the expression level of *PTGES* was a result of the inflammatory response, as lipopolysaccharides can induce the expression of *PTGES* in bovine mammary epithelial cells and cause reproductive dysfunction [32]. Kaewmala et al. [33] found the presence of the PTGS2 (COX-2) protein in boar Leydig cells, spermatogonia, and sperm cells, suggesting that it may play a role in spermatogenesis in pigs. Marques et al. [34] found two genes, *PLA2G4A* and *PTGS2*, in the testicular tissue of Yorkshire boars, and a gene *HPGDS* in the testicular tissue of Landrace pigs, which was involved in eicosanoid biosynthesis and arachidonic acid metabolism, whereas *PTGS2* and *HPGDS* are also involved in the cyclooxygenase pathway, and most of the genetic variation in semen traits is explained by different genes in these two strains. Its gene network analysis revealed that the candidate genes of the two strains were involved in a shared biological pathway in the mammalian testis, indicating that this pathway also had an important regulatory effect on testicular development.

The normal development of the testis is a prerequisite for spermatogenesis and effective fecundity in male mammals. Its development process is extremely delicate and complex, and is affected and regulated by many factors, but genetic regulation is dominant. The development of mammalian testis is first strictly regulated by a large number of protein-coding genes. In recent years, transcriptomics based on mRNA, miRNA, lncRNA and piRNA have been fully applied in the molecular mechanism of mammalian testicular development and spermatogenesis, and have made great progress. In this study, the early testicular development of piglets was studied using aspects of testicular development-related genes and metabolic pathways, to provide an important theoretical basis for further understanding the genomic changes of testicular development and animal breeding in animal husbandry. The molecular regulation of testicular development and spermatogenesis is crucial to improve the influence of postnatal boar fertility regulation, and is of great significance to improving the breeding efficiency and genetic resource conservation of the pig industry. However, these underlying regulatory activities and mechanisms require further study.

## 5. Conclusions

In this study, the genes *PTGES*, *SFRP1*, *SPP1*, *PLA2G4E*, *KCNJ5*, *PTGS2*, and *HCN1* that were involved in testicular development and spermatogenesis in boars were screened, and have provided a basis for the study of the molecular mechanism of early testicular development as well as data supporting the subsequent molecular-assisted breeding of breeding boars.

## Figures and Tables

**Figure 1 animals-14-00172-f001:**
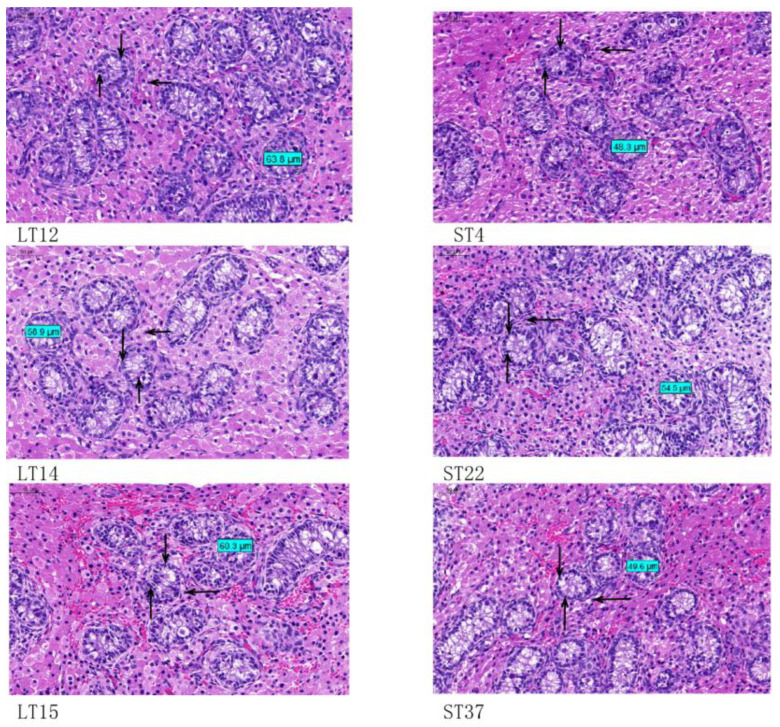
Histomorphology of large and small testis (400×). ↓: spermatogonia; ↑: Sertoli cells; ←: Leydig cells.

**Figure 2 animals-14-00172-f002:**
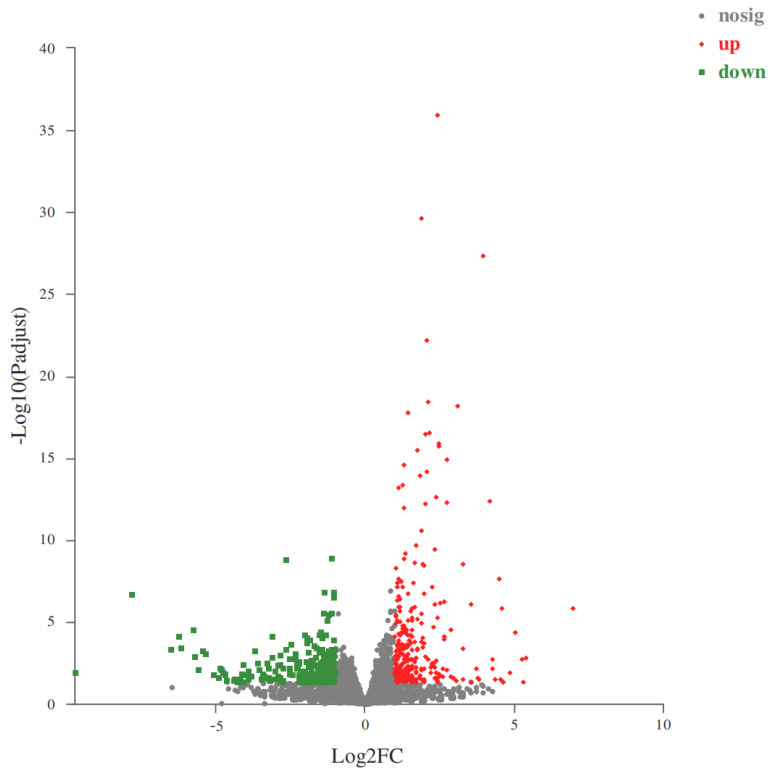
Volcano plot of differentially expressed genes in the testis of different sizes. The horizontal axis represents the multiple of gene expression differences between two samples, a numerical value obtained by dividing the expression level of the treated sample by the expression level of the control sample. The vertical axis represents the statistical test value of the difference in gene expression level, which is the *p*-value. The larger the log10 (*p*-value), the more significant the expression difference, and the values in the horizontal and vertical coordinates have been converted to logarithms. Each point in the graph represents a specific gene, with red dots representing significantly upregulated genes, green dots representing significantly downregulated genes and gray dots representing non-significantly differentially expressed genes. After mapping all genes, it can be seen that the points on the left are genes with downregulated expression differences and the points on the right are genes with upregulated expression differences. The closer the points are to the two sides and the top, the more significant the expression differences.

**Figure 3 animals-14-00172-f003:**
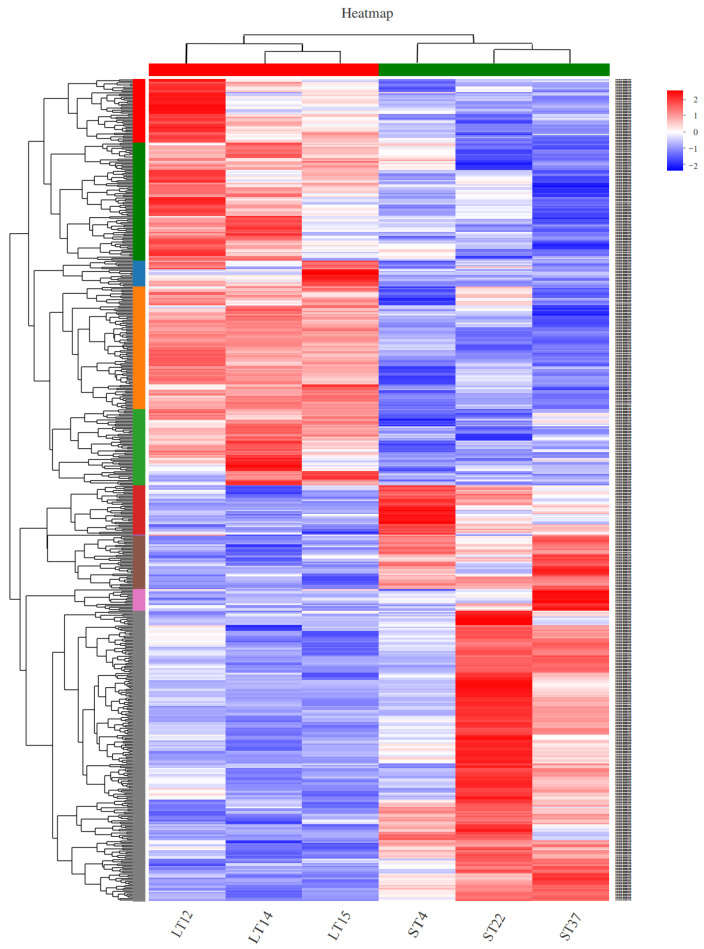
Heat map of DEGs in different sizes of testis. Each column in the graph represents a sample, and each row represents a gene. The colors in the graph represent the standardized expression levels of the gene in each sample, with red indicating a higher expression level of the gene in the sample and blue indicating a lower expression level. The specific trend of expression level changes can be found in the numerical annotation below the color bar in the upper left corner. The left side shows the tree diagram of gene clusters and the module diagram of sub clusters, while the right side shows the names of genes. The closer the two gene branches are, the closer their expression levels are. The upper part is a tree diagram of sample clustering and the lower part is the name of the sample. The closer the branches of the two samples are, the closer the trend in gene expression changes.

**Figure 4 animals-14-00172-f004:**
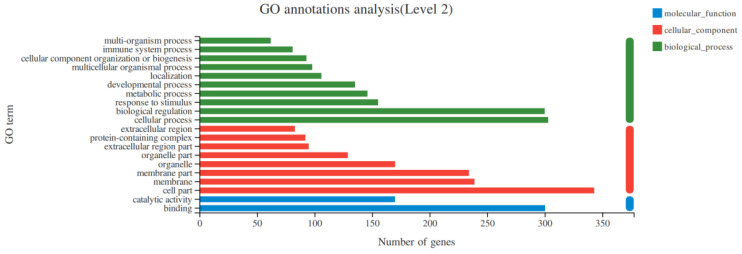
GO functional enrichment analysis of DEGs in different sizes of testes.

**Figure 5 animals-14-00172-f005:**
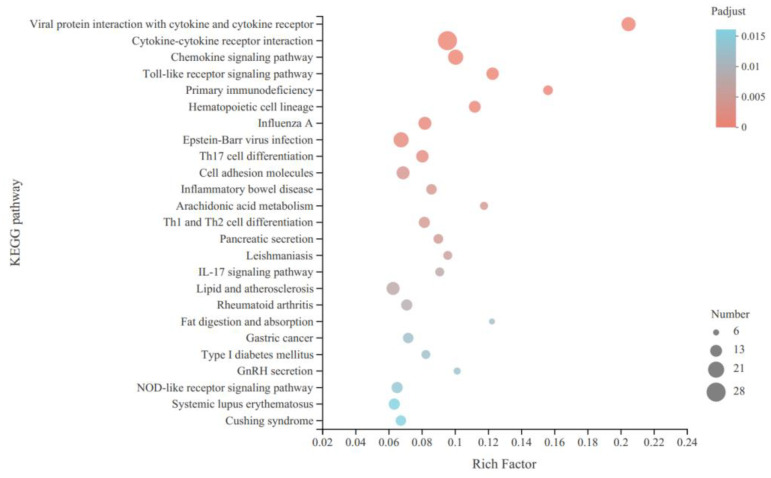
KEGG functional enrichment analysis of DEGs in the testes of different sizes.

**Figure 6 animals-14-00172-f006:**
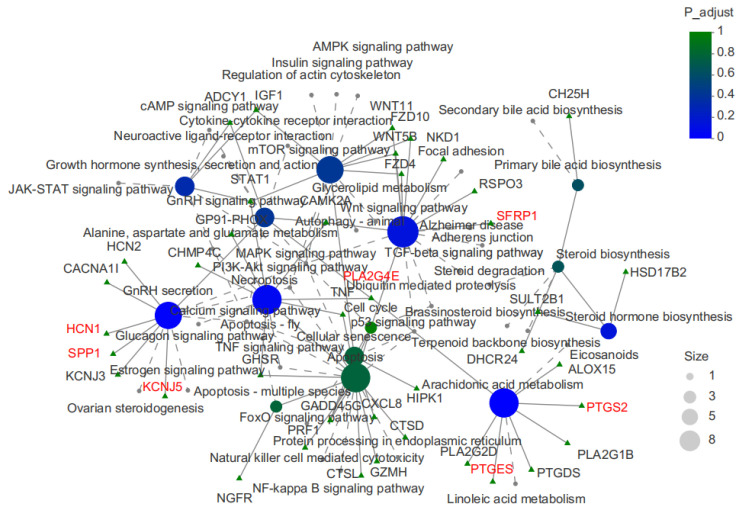
Network diagram of DEGs and KEGG pathway network in testes of different sizes. The green triangle nodes show genes; Circular nodes of different colors and sizes represent KEGG pathways, with color representing *p*-values and size representing the number of genes in the pathway. The selected candidate genes are indicated in red font.

**Figure 7 animals-14-00172-f007:**
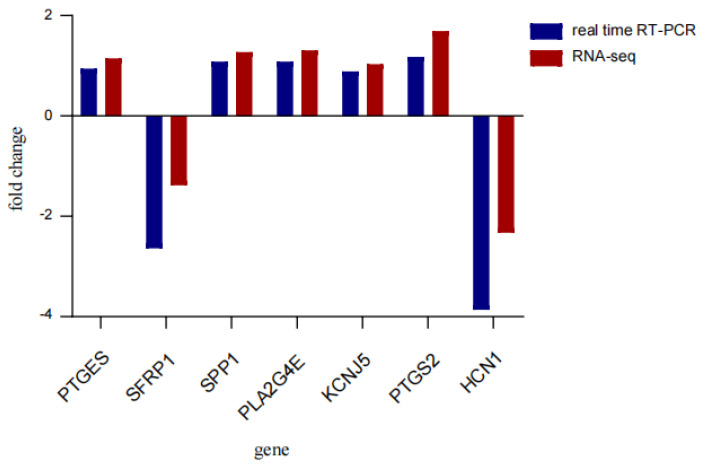
qRT-PCR validation of differential genes for different size testes.

**Table 1 animals-14-00172-t001:** PCR primer information for internal reference and differentially expressed genes.

Gene	Forward Primer (5′-3′)	Reverse Primer (5′-3′)
PTGES	AGTGAGGCTTCGGAAGAAGG	TCATTCCGATGGGCCCTAAG
SFRP1	GTGTCCTCCATGTGACAACG	CGATCTTCTTGTCGCCGTTT
SPP1	GCCACATTGCTAAAGCCTGA	TGGCAGGGTCTCTTGTTTGA
PLA2G4E	AATGTGATGCCAACGTCCTG	AGTTGTGGAACTGGGACACA
KCNJ5	TACCTGAGCGACCTCTTCAC	CCCGGATGTAAGCAATGAGC
PTGS2	TGATGGCCACGAGTACAACT	CTGGTCGATTGAGGCCTTTG
HCN1	CGAGAAGGAGCTGTGGGTAA	TCAGCAGGCAAATCTCTCCA
β-actin	CCCTGGAGAAGAGCTACGAG	TAGAGGTCCTTGCGGATGTC

**Table 2 animals-14-00172-t002:** Statistics of transcriptome alignment results of testis samples.

Sample	Total Reads	Total Mapped	Multiple Mapped	Uniquely Mapped
LT12	57,632,434	55,340,855 (96.02%)	1,873,579 (3.25%)	53,467,276 (92.77%)
LT14	62,742,476	60,345,536 (96.18%)	2,104,208 (3.35%)	58,241,328 (92.83%)
LT15	62,787,562	60,345,322 (96.11%)	1,982,702 (3.16%)	58,362,620 (92.95%)
ST4	61,715,704	59,368,830 (96.2%)	2,043,653 (3.31%)	57,325,177 (92.89%)
ST22	58,535,570	56,198,163 (96.01%)	1,814,622 (3.1%)	54,383,541 (92.91%)
ST37	57,182,910	55,152,747 (96.45%)	1,869,568 (3.27%)	53,283,179 (93.18%)

## Data Availability

The data presented in this study are available upon request from the corresponding author.

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
