# Peer review of "Identification and Analysis of Genes Related to Testicular Size in 14-Day-Old Piglets"

_animals, 2024, doi:10.3390/ani14010172_

Round 1

Reviewer 1 Report

Comments and Suggestions for Authors

In this study, RNA-Seq technology was used to screen the key genes affecting the early testicular development of Duroc × Landrace × Yorkshire piglets and determine their regulatory pathways, which provided a basis for exploring the molecular mechanism of early testicular development.

However, the article is not innovative enough, and the experimental data are less, the following questions are put forward:

1. Whether the ternary hybrid pig can be used as an animal model of gene expression regularity of testicular development.

2. The introduction of background is less and the structure level is not clear.

3. Article 2.1 does not explain the specific criteria for judging small testes and large testes.

4. "2.2. Materials and methods" is not specific enough. For example, please add or cite references to support specific steps for HE staining.

5. Figure 1 is not clear enough to determine cell diameter.

6. The conclusion is not concise enough. Just describe what the experiment found.

Reviewer 2 Report

Comments and Suggestions for Authors

The authors have tried to show the transcriptomic level changes in the early development genes in the boar utilizing bulk RNA-seq data. 

The bulk RNAseq data is not deposited in the GEO database. There is no information on the data and its availability. 

There is already Single-cell RNAseq data available in the GEO database which reveals the dynamic process and novel markers in porcine spermatogenesis. 

To make this study novel, the authors need to perform correlation studies to study the correlation of these early developmental diseases with these list of genes (PTGES, SFRP1, SPP1, PLA2G4E, KCNJ5, PTGS2, and HCN1). 

The authors should further correlate with the testicular biochemical, hormonal, and histological changes (they can calculate Testis weight, Intra-testicular testosterone, Insulin receptor, adiponectin receptors, germinal epithelial height, and tubular diameter) with the gene they are showing to make the paper interesting and novel (they can see and cite the article 30471430). 

The list of genes should be visualized in the heatmap, the position of all the genes in the heatmap.

Reviewer 3 Report

Comments and Suggestions for Authors

The authors conducted an interesting study on the genes expressed in 14-day-old pig according to their testis size. Considering that the study is not conducted over time, but in a single age, authors should change the title and provide another one that agrees with the research conducted. This is an important issue of this manuscript that should be addressed in the whole document.

Besides of this, all sections need a deep revision and improvement to provide a clear message. Much effort should be done to improve the Discussion by providing the relevance of the results obtained from a physiological point of view, instead of reviewing the role of the different genes. This section should also highlight the differences between large and small testes in terms of gene expression at this specific age (14-day-old).

All Figure legends need some revision and improvement, specifically Figures 1 and 3.

English grammar needs some revision and improvement.

Some specific comments are detailed below:

In lines 14-15 of the Simple Summary authors refer to “The plateau environment has an influence on the reproductive performance of boars and the testicular development of plateau bred piglets”. After reviewing the manuscript this reviewer is unable to know what do the authors refer to. So, they should review this section and provide a clear message that agrees with the research conducted.

Line 47. Please, revise and improve this sentence. It is not appropriate to refer to the testis as an important organ for producing sperm, considering that it is the only organ with this function.

Line 48. “Accommodating the growth and development of sperm cells”. This expression is not appropriate. Do the authors refer to spermatogenesis? If so, please use the appropriate terms.

Lines 48-50. “maintaining the normal progress of its physiological structure and physiological functions and has positive regulation of reproductive activities”. What do the authors refer? Please, note that this sentence is very ambiguous, and it does not provide a clear message.

Lines 52-53 “growth and function of sperm cells”. Again, the meaning of this expression is unclear. Sperm cells do not growth in the testis; besides testicular sperm are immature cells unable to fertilize the oocyte. According to this, authors should review and improve the sentence.

Lines 58-87. The text of these lines is bold highlighted; please, use normal text.

Line 63. Please, indicate the full name of lncRNAs.

Figure 1 contains low-quality histological images. Please, provide high quality images to better understand the testicular histology.

Line 97. Why did authors select only three testes per group for RNA-Seq? How many testicular samples were used for histological analysis? Considering that the number of piglets is 100, it is not clear how many testicular samples were used for the different experimental approaches.

Line 164. Please provide a detailed legend for Figure 1.

Line 190. Please provide a detailed legend for Figure 3.

Lines 252-256. Please, revise the message of these lines. I sincerely think that the study does not show that the testes with different size have different developmental paths and so different structure.

In general, the Discussion section is a review of the role of the different genes identified and it does not highlight the physiological relevance of the results obtained.

Line 292. What postnatal micro puberty is?

The Conclusion section also needs deep revision and improvement. Authors analysed the testes from 14-day-old piglets, so they can not conclude that “testicular development and spermatogenesis play a crucial role in the maturation of pigs” (lines 321-322).

In this section authors also refer to the “plateau environment” (line 328), but it is not clear what do the authors refer to. Did the authors really analyse it?

The Reference section also needs deep improvement. The reference list has a variable format, which does not agree with the journal recommendations.

Comments on the Quality of English Language

The text contain grammatical and typographycal errors that should be corrected during the reviewing process.

Reviewer 4 Report

Comments and Suggestions for Authors

In the article: “Identification and analysis of genes related to early testicular development in duroc × landrace × yorkshire piglets”, Zhao et al. using modern research methods, extracted total RNA, quantified gene expression, tested the accuracy of sequencing results to analyze the obtained data. For the study, they used testicles from 100, 14-day-old Duroc × Landrace × Yorkshire piglets of similar body weight and genetic origin, coming from the same breeding environment.

The topic of this work is original and relevant to the field, because researchers showed that the testicular development-related pathways identified in this study included arachidonic acid metabolism, Wnt signaling pathway and GnRH secretion, and seven candidate genes closely related to the development of testes of these piglets were screened, including PTGES, SFRP1, SPP1, PLA2G4E, KCNJ5, PTGS2, and HCN1.

 Compared to other published works, the authors' research reveals for the first time, based on high-throughput sequencing analysis and real-time fluorescence RT-PCR, that the mentioned genes may be involved in the regulation of germ cell development, spermatogenesis and semen characteristics. The authors obtained the results, thanks to which they opened a very interesting discussion, bringing a lot of new and interesting information closely related to the issues raised by them which may be a starting point for further research.

 In my opinion, the materials and research methods are adequate and do not raise any major objections. The conclusions are consistent with the evidence, and the arguments presented respond well to the main question posed. The thoroughly researched and up-to-date literature, with which the authors skillfully discuss, deserves praise. The value of the work increases especially thanks to the very well presented and described figures.

Here are my detailed comments and reservations:

The introduction does not clearly describe the purpose of the research.

Line 14: what is plateau bred piglets and also line 324: highland areas, could you explain what do you mean about these formulations more precisely?, line 328: the same – plateau environment?

Line 94: Please define the division into research groups more precisely, what separated them and what were the limit values that distinguished the two groups?

Line 96: what is the SC distribution?

Line 96-97: please provide cutoff values between groups.

Line 97: why ST group was the control and LT the experimental group?

Line 157: latency of what?

Line 231 and 366-367: Literature No. 10 should probably be in English, not Chinese?

Line 259: Suranai it is in literature item no. 20, not 18! The discussion seems to refer to Wong et al. in citation position no. 20?

Line 326: This provides… what? Could you explain the meaning of this sentence in more detail?

Round 2

Reviewer 2 Report

Comments and Suggestions for Authors

The author's comments are not convincing. The authors should try to mark the possible and answer the comments in a point. It's not our responsibility to search the response to reviewer comments. 

Also, I suggest authors cite article 30471430 and correlate it with the testicular biochemical, hormonal, and histological changes (they can calculate Testis weight, Intra-testicular testosterone, Insulin receptor, adiponectin receptors, germinal epithelial height, and tubular diameter) with the gene they are showing to make the paper interesting and novel. The authors can easily calculate the histological parameter using the article. 

Author Response

Dear Reviewers,

We have made revisions to your suggestions in the article and highlighted them in a revision mode. However, we are unable to include the calculation of histological parameters. This experiment is a transcriptomic analysis designed around a single factor. Detailed histological discussions will be another topic, and physiological and biochemical data for this batch of samples cannot be obtained at present. We apologize for not being able to add this part of the content. Our research depth in organizational science is not high, and I still need to learn this knowledge. Thank you for your suggestions, which have brought us improvement. The progress of this work is quite difficult.

We would like to take this opportunity to thank you for all your time and for the excellent opportunity you have given us to improve the manuscript. We hope you will find this revised version satisfactory.

Sincerely

Author

Reviewer 3 Report

Comments and Suggestions for Authors

Authors introduced some of the modifications requested by this reviewer; nevertheless, there are also several issues that still needs to be addressed.

English grammar still needs revision and improvement.

Line 49. “The testis is an important organ for producing sperm”. As I already commented in the previous revision this sentence needs some revision and improvement.

Lines 95-96. Again, the authors did not study “the temporal changes of testicular gene expression during fetal testicular development”. According to this, the sentence should be modified.

Lines 206-207. “The number of supporting cells is related to the size of the testis and also determines the ability of spermatogonia to proliferate and differentiate”. This sentence is not derived from the research conducted. So, authors should add a bibliographic reference.

Line 214. Please, note that this “supporting cells” are named “Sertoli cells”.

Lines 223-224. Please, provide an appropriate legend for Figure 1. This is modifications was already asked in the previous revision. Authors should indicate the histological structures and cell types present in the histological images and indicate which of the refer to large and small testes, and which are the main differences between them. Moreover, the pictures have a code number; which is the meaning of LT12, LT14 and LT15? And of ST$, ST22 and ST37?

Line 232. In the Sample column of this table, the samples’ name is either LT or ST. Are these names related to the differences in sample size? If so, please, specify in a table foot.

Line 240. Please, provide an appropriate legend for Figure 2. Why the variables of X and Y axes are Log2FC and Log10P adjust. Please, provided an adequate explanation in the Material and Methods section. Explain the meaning of these variables in the figure legend.

Line 249. Please, provide an appropriate legend for Figure 3.

Line 287. Please, provide an appropriate legend for Figure 7. Authors should indicate ethe meaning of the abbreviations in X axis.

Line 311-312. “2-3 weeks after birth is the peak period for the proliferation of testicular interstitial cells in piglets.” Please, add a bibliographic reference at the end of this sentence.

Line 365. The term “postnatal micro-puberty” is also present in the revision version of the manuscript. Again, I ask the authors to clearly indicate what do they refer to.

Line 388-389. “effective fecundity”. Again, I do not understand what do the authors refer to. Even, I am not sure if this term is appropriate in the context of the research conducted.

Comments on the Quality of English Language

English grammar still needs some revision and improvement,

Author Response

Dear Reviewers,

We have made revisions to your suggestions in the article and highlighted them in a revision mode. Regarding the group names appearing in the results section, the materials and methods indicate that ST represents the small testicular group, LT represents the large testicular group, and numbers represent the sampling order. Changes will disrupt a large amount of other data modifications, so no modifications have been made. In Figure 7: the X axis shows the gene names, and the Y axis shows the expression of each gene.

Thank you for the detailed review. We have carefully and thoroughly proofread the manuscript to correct all the grammar and typos.

We would like to take this opportunity to thank you for all your time involved and this great opportunity for us to improve the manuscript. We hope you will find this revised version satisfactory.

Sincerely,

The Authors

Reviewer 4 Report

Comments and Suggestions for Authors

Thank you for taking into account my comments and the suggestions of other reviewers. In its current form, the work meets the submitted requirements.

Author Response

Dear Reviewers,

    Thank you for your comments and suggestions.

Sincerely,

The Authors